# Bistabilities and domain walls in weakly open quantum systems

**Florian Lange and Achim Rosch**[⋆]

Institute for Theoretical Physics, University of Cologne,
Zülpicher Straße 77a, D-50937 Cologne, Germany

⋆ rosch@thp.uni-koeln.de

## Abstract

Weakly pumped systems with approximate conservation laws can be efficiently described by (generalized) Gibbs ensembles if the steady state of the system is unique. However, such a description can fail if there are multiple steady state solutions, for example, a bistability. In this case domains and domain walls may form. In one-dimensional (1D) systems any type of noise (thermal or non-thermal) will in general lead to a proliferation of such domains. We study this physics in a 1D spin chain with two approximate conservation laws, energy and the $z$-component of the total magnetization. A bistability in the magnetization is induced by the coupling to suitably chosen Lindblad operators. We analyze the theory for a weak coupling strength $\epsilon$ to the non-equilibrium bath. In this limit, we argue that one can use hydrodynamic approximations which describe the system locally in terms of space- and time-dependent Lagrange parameters. Here noise terms enforce the creation of domains, where the typical width of a domain wall goes as $\sim 1/\sqrt{\epsilon}$ while the density of domain walls is exponentially small in $1/\sqrt{\epsilon}$. This is shown by numerical simulations of a simplified hydrodynamic equation in the presence of noise.

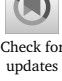

# 1  Introduction

In the thermodynamic limit the steady state of an interacting many-body quantum system can be described in a very compact way by a Gibbs ensemble, $\rho \sim e^{-\sum_{i=1}^{N} \lambda_i Q_i}$, where the $Q_i$ are $N$ conserved quantities (energy, particle number, magnetization, ...) of the system. Here the Lagrange parameters $\lambda_i$ are in one-to-one relation to the expectation values of the $Q_i$. In one-dimensional integrable many-particle systems $N$ grows linearly with system size, in this case the term 'generalized Gibbs ensemble' is used [1–3]. This approach can even be used if the conservation laws are only approximately valid and if the system is weakly driven out of equilibrium as long as scattering processes which conserve the $Q_i$ dominate the dynamics. For example, to describe the Bose-Einstein condensation of exciton-polaritons or photons [4–8], it is useful to introduce a chemical potential for these particles despite the fact that particle number is not exactly conserved in the systems. The value of the chemical potential is then determined by balancing loss and pumping rates. Similarly, in solid state materials driven out of equilibrium, e.g., by a short laser pulse, one can use the weak coupling of phonons to electrons to introduce two different temperatures for the two subsystems. Here the relevant approximately conserved quantities $Q_1$ and $Q_2$ are the energies of the phonon and electron system, respectively. The corresponding $\lambda_i$ are identified with their inverse temperatures. Simple rate equations then describe the time-evolution within such two-temperature models [9]. Recently, we have generalized this notion also to approximately integrable systems with

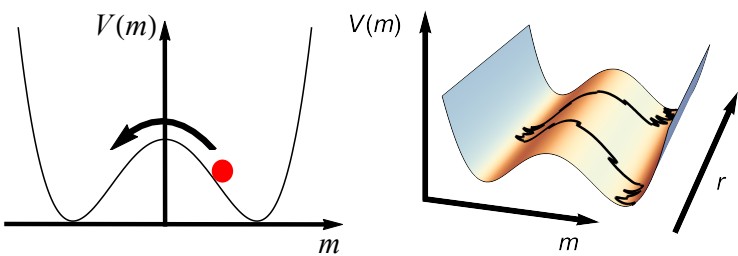

Figure 1: Left: Single particle in a 0D system whose deterministic dynamics is governed by a symmetric double-well potential. In the absence of noise, the symmetry of the potential is broken by the initial conditions. However, any finite noise triggers consecutive transitions between the two minima of the potential and eventually restores the symmetry in the long-time limit. Right: A 1D generalization of the aforementioned case for a field theory. Similarly to the 0D case there is no symmetry breaking at finite noise strength. Instead, depending on the noise strength a non-zero density of domain walls forms.

an infinite number of – approximate – conservation laws, where we could show that one can create giant heat and spin currents in driven spin chains [10]. Similar concepts can also be used to describe many-body localized systems coupled to phonons and an external drive [11]. In this work, we want to study in a controlled way a weakly driven system with approximate conservation laws where the concept of a (generalized) Gibbs ensemble breaks down. Starting from a 1D system with just two exact conservation laws (energy and magnetization), we add weak perturbations of strength $\epsilon$ which break the corresponding symmetries and drive the system out of equilibrium. We choose the perturbations in such a way that they induce a bistability in the magnetization and argue that noise terms naturally generate domains and domain walls in such systems. The emergence of bistabilites in non-equilibrium systems in the

limit of strong drive and dissipation has also gained increased attention due to experimental observations in a wide range of systems, including, for example, driven Rydberg ensembles [12,13], nonlinear photon lattices [14–16], semiconductor microcavities [17], and QED setups with cold atoms [18,19].

In low-dimensions a symmetry cannot be spontaneously broken due to thermal fluctuations. A well known zero-dimensional example is the supercritical pitchfork bifurcation [20] with additive noise $\xi(t)$, i.e. $\dot{x}(t) = -V'(x) + \sqrt{2\alpha}\xi(t)$, $V(x) = -\frac{\mu}{2}x^2 + \frac{1}{4}x^4$, $\langle \xi(t)\xi(t')\rangle = \delta(t-t')$. For the deterministic part of the dynamics ($\alpha = 0$) one obtains for $\mu > 0$ two stable solutions at $\pm\sqrt{\mu}$, see Fig. 1. In the absence of noise, the symmetry of the underlying symmetric double-well potential is broken by the initial conditions. However, at any finite noise strength $\alpha \neq 0$, the symmetry is restored in the long-time limit and the corresponding Focker-Planck equation yields $P(x) \sim \exp(-V(x)/\alpha)$ as a stationary probability distribution. Ref. [21] discusses, for example, such a zero-dimensional case by investigating a Dicke model with non-linear noise. Similarly, in 1D systems with short-ranged interactions arbitrarily weak noise will generically induce domain walls thus rendering any description in terms of noiseless (generalized) Gibbs ensembles invalid. Here the finite cost of a domain wall plays a similar role as the potential barrier of the zero-dimensional example, see Fig. 1. In dimensions larger than one, in contrast, an Ising symmetry can be spontaneously broken even in the presence of (sufficiently weak) noise [22, 23]. In the following, we will investigate a simple 1D model which allows one to study the role of approximate conservation laws, the validity of Gibbs ensembles and the relations of bistabilities and noise in a controlled way. We discuss how an effective description in terms of (noisy) hydrodynamics can be obtained and solve a simplified version of these equations numerically.

## 2  Model

We consider a antiferromagnetic ($J > 0$) one-dimensional XXZ spin chain

$$H_0 = J\sum_j \left(\sigma_j^+\sigma_{j+1}^- + \sigma_j^-\sigma_{j+1}^+ + \Delta\sigma_j^z\sigma_{j+1}^z\right) + J'\sum_j \left(\sigma_j^+\sigma_{j+2}^- + \sigma_j^-\sigma_{j+2}^+\right).$$

The next-nearest neighbor interaction $J'$ renders the model non-integrable. The unperturbed Hamiltonian $H_0$ therefore has only two conservation laws: the magnetization in $z$-direction and the energy, $[H_0, Q_i] = 0$ with

$$Q_1 = \sum_j \sigma_j^z, \qquad Q_2 = H_0. \tag{1}$$

We assume that the system is driven out of equilibrium by the weak coupling to a Markovian bath. The dynamics of the density matrix $\rho$ is thereby governed by the Liouville equation

$$\dot{\rho} = \hat{\mathcal{L}}\rho = \left(\hat{\mathcal{L}}_0 + \epsilon\hat{\mathcal{L}}_1\right)\rho, \tag{2}$$

$$\hat{\mathcal{L}}_0\rho = -i[H_0, \rho], \quad \hat{\mathcal{L}}_1\rho = (1-\gamma)\hat{\mathcal{D}}_1\rho + \gamma\hat{\mathcal{D}}_2\rho, \tag{3}$$

$$\hat{\mathcal{D}}_i\rho = J\sum_j L_j^{(i)}\rho L_j^{(i)\dagger} - \frac{1}{2}\{L_j^{(i)\dagger}L_j^{(i)}, \rho\}, \tag{4}$$

where, importantly, $\epsilon$ is assumed to be small. We aim to construct the coupling such that the dynamics exhibits a (local) bistability in the presence of noise. To achieve this goal we consider two competing perturbations $\hat{\mathcal{D}}_i$ (i=1,2) whose relative strength is controlled by the

parameter $\gamma \in [0,1]$. As Lindblad operators we choose

$$L_j^{(1)} = \sigma_j^x, \tag{5}$$

$$L_j^{(2)} = P_{j-1}^{\uparrow} \sigma_j^+ P_{j+1}^{\uparrow} + P_{j-1}^{\downarrow} \sigma_j^- P_{j+1}^{\downarrow}. \tag{6}$$

While the first Lindblad operator flips spins which leads to noise and heating, the second Lindblad operator aligns neighboring spins by transforming ↑↓↑ to ↑↑↑ and ↓↑↓ to ↓↓↓ ($P_j^{\uparrow/\downarrow} = \frac{1}{2}(1 \pm \sigma_j^z)$ are projection operators on up/down spin configurations, respectively). Therefore it naturally induces a bistability in the total magnetization of the system. For $\gamma = 1$, i.e., in the absence of the $\sigma^x$ term, the fully polarized states $|\Uparrow\rangle = |\uparrow \dots \uparrow\rangle$ and $|\Downarrow\rangle = |\downarrow \dots \downarrow\rangle$ are the two unique dark states of the system and the steady-state density matrix is simply given by $\rho = p|\Uparrow\rangle\langle\Uparrow| + (1-p)|\Uparrow\rangle\langle\Uparrow| + (\alpha|\Uparrow\rangle\langle\Downarrow| + h.c.)$, describing a state with spontaneously broken symmetry. The existence of unique dark states in a many-body system is, however, not the generic case. In the following we will only consider the situation where such dark states do not exist, i.e., we consider the case $\gamma < 1$ only.

## 3 Hydronicamic Approximations

For $\epsilon = 0$, in the absence of any coupling to an environment, the steady-state density matrix in the thermodynamic limit is simply given by $\rho \sim e^{-\lambda_1 Q_1 - \lambda_2 Q_2}$. Here the parameters $\lambda_1$ and $\lambda_2$ are not fixed by the dynamics but only by initial conditions. Scattering processes of the non-integrable interacting system are essential to establish this steady state. For a finite, but tiny value of $\epsilon$ it is clear that the system will remain locally close to such states (for a quantitative discussion of corrections we refer to Ref. [24]). If such stationary states are not unique (e.g, due to a bistability), we can, however, *not* expect that locally the same values of $\lambda_i$ are obtained as we will show in detail below. Instead, we should parametrize the system with space-dependent Lagrange parameters $\lambda_i(r)$. This leads to the following ansatz for the density matrix

$$\rho \approx \int \mathcal{D}[\lambda_i(r)] \, P_t[\lambda_i(r)] \left( \frac{e^{-\int dr \sum_{j=1,2} \lambda_j(r) \hat{q}_j(r)}}{Z[\lambda_i]} + \delta\rho \right). \tag{7}$$

Here we integrate (in the functional integral sense) over smoothly varying space-dependent Lagrange parameters $\lambda_i(r)$. The $\hat{q}_i(r)$ are the (coarse-grained) local charge density operators with $Q_i = \int dr \, \hat{q}_i(r)$, $Z[\lambda_i]$ is the partition sum for a fixed configuration of $\lambda_i(r)$, and $\delta\rho$ is a correction to the density matrix arising from gradients of $\lambda_i(r)$, briefly discussed below. The (yet unknown) functional $P_t[\lambda_i(r)]$ describes the (classical) probability for a given configuration of Lagrange parameters defined by $\lambda_1(r)$ and $\lambda_2(r)$. In general, $P_t[\lambda_i(r)]$ depends on time. It describes the dynamics on a time scale of order $1/\epsilon$, which is assumed to be much larger than all internal equilibration times [25]. Instead of developing directly a theory for the probability distribution $P_t[\lambda_i(r)]$ in the spirit of a Fokker-Planck equation, we will use a description in terms of a (generalized) Langevin equation for the fields $\lambda_i(r)$ or, equivalently, the corresponding local expectation values of the charge densities $\hat{q}_i(r)$. This approach has the advantage of being much more intuitive. In the following we use $q_i(r,t)$ to denote the expectation values of the coarse-grained local densities for *one* realization of the underlying Langevin process.

Technically, we will perform a gradient expansion around the homogeneous solutions $q_i(r) = const$ [26,27]. To zeroth order in the gradient expansion, we can locally approxi-

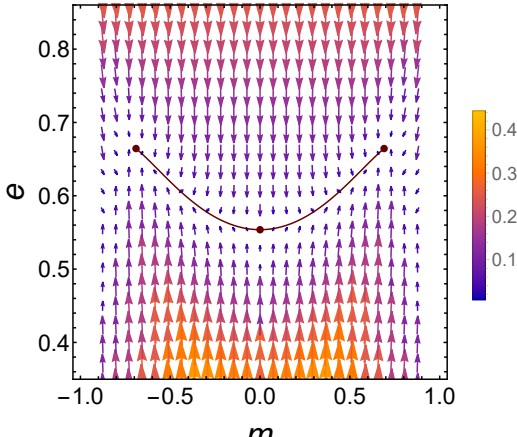

Figure 2: Plot of the force field $(f_1, f_2)$ defined in Eq. (9) as a function of the magnetization $q_1 = m$ and the energy density $q_2 = e$. The color encodes the amplitude of the force. The force vanishes at two stable fixed points with a finite magnetization $m \approx \pm 0.7$ and at an unstable fixpoint at $m = 0$ (red points). The red solid line indicates the pathes from the unstable to the stable fixed points. Parameters: $N = 12, \gamma = 0.9, J = \Delta = 1, J' = 0.01$.

mate the density matrix close to the position $r_0$ by

$$\rho_{r_0}^{(0)}(t) \approx \frac{e^{-\sum_j \lambda_j(r_0,t)Q_j}}{\mathrm{Tr}[e^{-\sum_j \lambda_j(r_0,t)Q_j}]}. \tag{8}$$

We use this density matrix to compute the change of the conserved charge densities linear in $\epsilon$

$$\partial_t \left\langle \frac{Q_i}{L} \right\rangle_{r_0}^{(0)} = \epsilon \mathrm{Tr}\left[ \frac{Q_i}{L} \hat{\mathcal{L}}_1 \rho_{r_0}^{(0)}(t) \right] = f_i(q_1(r_0), q_2(r_0)). \tag{9}$$

Here $L$ is the size of the system and $f_i$ is the averaged, deterministic force which depends on the local Lagrange parameters $\lambda_i(r_0)$, or equivalently on the local densities $q_i(r_0)$ ($i = 1, 2$) evaluated at $r_0$. Within our Langevin approach we expect that the coupling to the bath also leads to a noise term $\xi_i(r)$,

$$\partial_t q_i(r, t) = f_i(q_1(r, t), q_2(r, t)) + \xi_i(r, t), \tag{10}$$

with $\langle \xi_i \rangle = 0$ and $\langle \xi_i(r, t)\xi_j(r', t') \rangle \approx N_{ij}\delta(t - t')\delta(r - r')$. As we describe in Appendix A.2, the noise matrix $N_{ij}$ can be computed from the time evolution of $Q_iQ_j$ [28]. Importantly, both the forces $f_i$ and the noise matrix $N_{ij}$ are linear in $\epsilon$ as they arise both from the coupling to the Lindblad operators. Both are also functions of the local charges $q_i(r, t)$. We compute $f_i$ and $N_{ij}$ using exact diagonalization of small systems. For the parameters investigated by us the effective temperatures are rather high and thus finite size effects turn out to be small. Fig. 2 displays the forces for $\gamma = 0.9$. As expected from our construction, we obtain two stable fixed points at a magnetization of approximately $m \approx \pm 0.7$. We denote the values of conservation laws at the fixed points as $q_i^{FP1}$ and $q_i^{FP2}$ with $q_1^{FP1} = -q_1^{FP2}$ and $q_2^{FP1} = q_2^{FP2}$ by symmetry. In the absence of the noise term, these two solutions would lead to spontaneous symmetry breaking. We also find an unstable fixed point at $m = 0$ and $e \approx 0.55$. It is important to note that in the presence of approximate conservation laws even a tiny coupling to a non-equilibrium bath

can strongly modify the system. In our example a state with a large magnetization and high energy is approached in the long-time limit even for very small perturbations $\epsilon$.

As a next step, we have to compute contributions to $\partial_t q_i$ arising from terms proportional to gradients of the local charges $\partial_r q_i$. Due to the space-reflection symmetries all linear gradients vanish on average. The other gradient terms can be calculated for $\epsilon = 0$ as they are finite in this limit. Their form is well known from standard hydrodynamics [26, 27] and we obtain

$$\partial_t q_i - \sum_j \partial_r D_{ij} \partial_r q_j = f_i + \xi_i + \partial_r \xi_i^{th}. \tag{11}$$

Here, $D_{ij}$ is the matrix of diffusion constants of the unperturbed model $H_0$ defined by $j_i = -D_{ij}\partial_r q_j$ where $j_i$ is the current of the conserved densities $q_i$. Technically, they arise from the correction $\delta\rho$ in Eq. (7) which induces gradients of the Lagrange parameters. The matrix of diffusion constants depends on $q_1$ and $q_2$ and can at $\epsilon = 0$ be computed using Kubo formulas evaluated in the corresponding thermal Gibbs state. The first two terms on the right-hand side have been copied from Eq. (10). The last term, again computed for $\epsilon = 0$, is the usual thermal noise with

$$\langle \xi_i^{th}(r,t)\xi_j^{th}(r',t')\rangle \approx \left(D_{ik}\chi_{kj} + \chi_{ik}D_{kj}\right) T \delta(t-t')\delta(r-r'), \tag{12}$$

where $\chi_{ij} = \frac{1}{TL}(\langle Q_i Q_j\rangle - \langle Q_i\rangle\langle Q_j\rangle)$ are the susceptibilities of the $Q_i$. Note that the thermal noise $\partial_r \xi_i^{th}$ obeys the conservation laws as it is proportional to a derivative while the non-equilibrium noise $\xi_i$ does not. The equations (11) describe the hydrodynamics of our driven system and we expect that they are exact in the limit of small $\epsilon$ as they have been derived in a systematic expansion in $\epsilon$ and gradients, keeping always the leading corrections. To understand their properties in the limit of small $\epsilon$, it is useful to rewrite the equations using rescaled variables. Employing that the forces are linear in $\epsilon$, we introduce rescaled variables, $\tau = t\epsilon$, $x = r\sqrt{\epsilon}$, $\tilde{f} = f/\epsilon$, $\tilde{\xi} = \xi/\epsilon$, $\tilde{\xi}^{th} = \xi^{th}/\sqrt{\epsilon}$. In these variables, we obtain equations which have exactly the same form as Eqs. (11),

$$\partial_\tau q_i - \sum_j \partial_x D_{ij} \partial_x q_j = \tilde{f}_i + \tilde{\xi}_i + \partial_x \tilde{\xi}_i^{th}. \tag{13}$$

The only difference is that now $\tilde{f}$ is *independent* of $\epsilon$ and the only $\epsilon$ dependence arises from the two noise terms which both turn out to be proportional to $\sqrt{\epsilon}$,

$$\langle \tilde{\xi}_i \tilde{\xi}_j\rangle \propto \sqrt{\epsilon}\delta(\tau-\tau')\delta(x-x'),$$
$$\langle \tilde{\xi}_i^{th}\tilde{\xi}_j^{th}\rangle \propto \sqrt{\epsilon}\delta(\tau-\tau')\delta(x-x'). \tag{14}$$

This immediately shows that both noise terms are of equal importance for our hydrodynamic theory. Furthermore, the analysis justifies a posteriori the gradient expansion underlying the derivation of our equation: higher order terms are suppressed by powers of $\sqrt{\epsilon}$. All parameters of our hydrodynamic equations can in principle be calculated from correlation functions of the unperturbed system $H_0$ only. By far the most difficult part of the calculation is the numerical determination of the diffusion constants $D_{ij}$ of the unperturbed system as function of the $q_i$. While there has been an enormous recent progress in the numerical calculation of transport coefficients in 1D systems [29], this is still a challenging problem suffering from huge finite size effects. As all of our qualitative results do not depend on the numerical values and functional dependence of the transport coefficients, we are not trying to calculate those. Instead, we will use in the following mainly the scaling arguments given above in combination with a numerical investigation of a strongly simplified version of Eqs. (11).

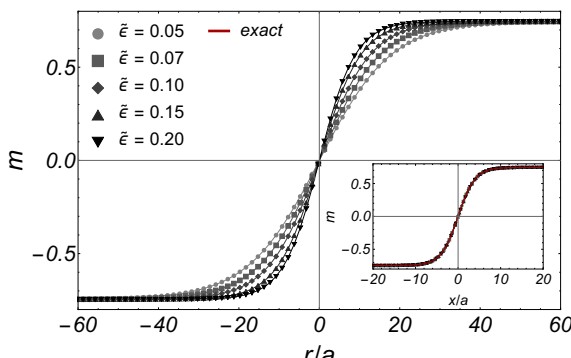

Figure 3: Steady state configuration of a domain wall in the absence of noise for different values of $\tilde{\epsilon} = (Ja^2/D)\epsilon$ as a function of position. The inset uses rescaled coordinates $x/a$ with $r = x/\tilde{\epsilon}$ and shows that the numerical results are well described by the solution of the corresponding field theory (red solid line), Eq. (16).

## 4 Simplified hydrodynamic model: order parameter theory

To obtain a simplified version of Eqs. (11) we proceed in the following way: First, instead of tracking the dynamics in the two-dimensional space $q_1$ and $q_2$, we concentrate on the magnetization $m$ as this is the only variable which shows a bifurcation and thus the order parameter of the model. Second, we replace the $q_i$ dependent matrix of diffusion coefficients by a single constant $D$. Finally, we adjust the forces of the right-hand side accordingly and obtain a strongly simplified model for the fluctuation induced domain-wall formation

$$\partial_t m - D\partial_r^2 m = f(m) + \xi + \partial_r \xi^{th}. \tag{15}$$

As we are only interested in the $\epsilon$ dependence of our result, we approximate the force by $f(m) = \epsilon J \left( \frac{\gamma}{4}(1-m^2) - (1-\gamma) \right) m$, set $\langle \xi\xi \rangle = 4aJ\epsilon(1-\gamma)\delta(r-r')\delta(t-t')$ ($a$ is the lattice constant of the microscopic model), and $\langle \xi^{th}\xi^{th} \rangle = 2D\chi T\delta(r-r')\delta(t-t')$, where we simply set $\chi T = a/2$ for our toy model. The functional form used for $f$ and the non-thermal noise is motivated by the infinite temperature limit where one can easily calculate all terms analytically, see App. A.1 and App. A.2. Within our toy model, a bistability is obtained for $\gamma > 4/5$ in the noiseless case.

Formally, the use of the simplified hydrodynamic theory with only a single mode, the order parameter of the bistabilty, is justified by the main goal of our study: we want to obtain the *qualitative* properties of the bistable system in the limit of small $\epsilon$. While the focus on just the order parameter is a well established approximation in equilibrium systems, it is necessary to revisit the argument in a non-equilibrium situation where static and dynamic properties might get mixed in a different way. For our argument we consider the rescaled theory (13). First, for $\epsilon = 0$ in the rescaled theory (note that this is different from the $\epsilon \to 0$ limit of the initial problem), all noise terms are absent and both the model (13) and the order parameter theory (15) exhibit Ising-type ferromagnetic order and in both theories the same type of domain walls (see discussion below) with the same scaling properties exist. Most importantly, all static and dynamical correlation functions of $q_1(x,t)$ and $m(x,t)$ evaluated at $\epsilon = 0$ have the same scaling properties in the two models. Furthermore, no qualitative changes can arise from the energy mode, $q_2$, as it obtains a finite mass which is of order 1 in the rescaled theory. This mass is simply given by $-\frac{d\tilde{f}_2}{dq_2}$ and describes physically that due to the coupling to the bath the energy relaxes to its steady state value. Omitting such a massive mode will not affect any scaling properties. Also the omission of nonlinear corrections arising from the $q_i$ dependence of diffusion constants is not expected to induce any qualitative changes as the theory retains

strong non-linearities (of order 1 in the rescaled theory) from $f(m)$. In conclusion, we can expect that for small $\epsilon$ all scaling properties as function of $\epsilon$ remain identical for the full and the simplified hydrodynamic theory.

To analyze the properties of Eqs. (11) (and its simplified version Eq. (15)), we first consider the noiseless limit by neglecting $\xi$ and $\xi^{th}$. In this case, two trivial solutions are given by the fixed points, $q_i = q_i^{FP1}$ and $q_i = q_i^{FP2}$. More importantly, there is also a 'domain wall' solution obtained from the boundary condition $\lim_{r\to-\infty} q_i = q_i^{FP1}$ and $\lim_{r\to\infty} q_i = q_i^{FP2}$. As it is obvious from our scaling analysis, the width of the domain wall is proportional to $1/\sqrt{\epsilon}$. For our toy model, one can calculate the shape of the domain wall also analytically, by solving the static and noiseless version of differential equation Eq. (15) given by $D\partial_r^2 m = -f(m)$ with the boundary condition $\lim_{r\to\pm\infty} m(r) = \pm m_0$ which gives

$$m(r) = m_0 \tanh\left[\frac{r}{x_0/\sqrt{\epsilon}}\right], \tag{16}$$

with $x_0 = \sqrt{\frac{8D}{(5\gamma-4)J}}$ and $m_0 = \sqrt{5-4/\gamma}$. While such an analytic solution can only be obtained for the simplified model (15), we would like to emphasize that a very similar domain wall also has to exist in the hydrodynamic theory of the original model, Eq. (11). Both the presence of energy diffusion and non-linearities in the matrix of diffusion constants will change the precise shape of the domain wall but will not modify the scaling of its width with $1/\sqrt{\epsilon}$.

Fig. 3 shows such a domain wall for the toy model where it is compared to our numerical results. In our numerical simulations we use rescaled variable where length and time are measured in units of $a$ and $a^2/D$, respectively. Equivalently, one can set $D = J = a = 1$ and replace $\epsilon$ by $\tilde{\epsilon} = \epsilon(Ja^2/D)$. We discretize space in steps of size 0.25 and time in steps of 0.001, using Heun's method for integration [30].

As discussed in the introduction, we expect that for any finite strength of fluctuations, a finite density of such domain walls occurs in the steady state. This is confirmed by simulations of our simplified model, Eq. (15), shown in Fig. 4 for different values of $\epsilon$. The figure shows $m(r, t)$ after some initial waiting time in which the system obtains its (fluctuating) steady state.

For $\tilde{\epsilon} = 0.1$, domains are huge but their size drops rapidly when $\tilde{\epsilon}$ is increased. The time scale which governs a reversal of the local magnetization depends also strongly on $\epsilon$. In Fig. 5 we show the density of domain walls, or equivalently, the inverse distance of domain walls obtained for the model Eq. (15) which includes two types of noise terms.

Interestingly, one can obtain the density of domain walls analytically if one neglects the thermal fluctuations $\xi^{th}$. In this case it turns out that one can use well-known results obtained for equilibrium systems. Here it is important to note that our effective theories Eqs. (11) and also Eq. (15) are *not* equivalent to an equilibrium theory (they will, for example, not fulfill the second law of thermodynamics) as the two noise terms do not encode thermal noise of a single temperature. If we, however, switch off the noise contribution $\xi^{th}$ in Eq. (15), the resulting equation is equivalent to the dynamics of a non-conserved Ising order parameter dominated by friction (model A in the classification scheme of Halperin and Hohenberg) in the presence of thermal fluctuations.

The Ginzburg-Landau theory of the corresponding field theory is given by $\frac{1}{a}\int \frac{D}{2}(\nabla m)^2 + v(m)$ with $v(m) = -\int_0^m f(m')dm'$. The prefactor $1/a$ has been chosen such that units of energy are obtained. In these units the friction coefficient is set to $1/a$. Within this theory, the energy $E_{DW}$ of a domain wall is proportional to $\sqrt{\epsilon}$ or more precisely

$$E_{DW} = \frac{\sqrt{2JD(5\gamma-4)^3}}{3a\gamma}\sqrt{\epsilon}. \tag{17}$$

The effective temperature $T_{eff} = \epsilon J(1-\gamma)$ is set by the strength of fluctuations of $\xi$ and therefore linear in $\epsilon$. Hence, we expect that the density of domain walls is proportional to

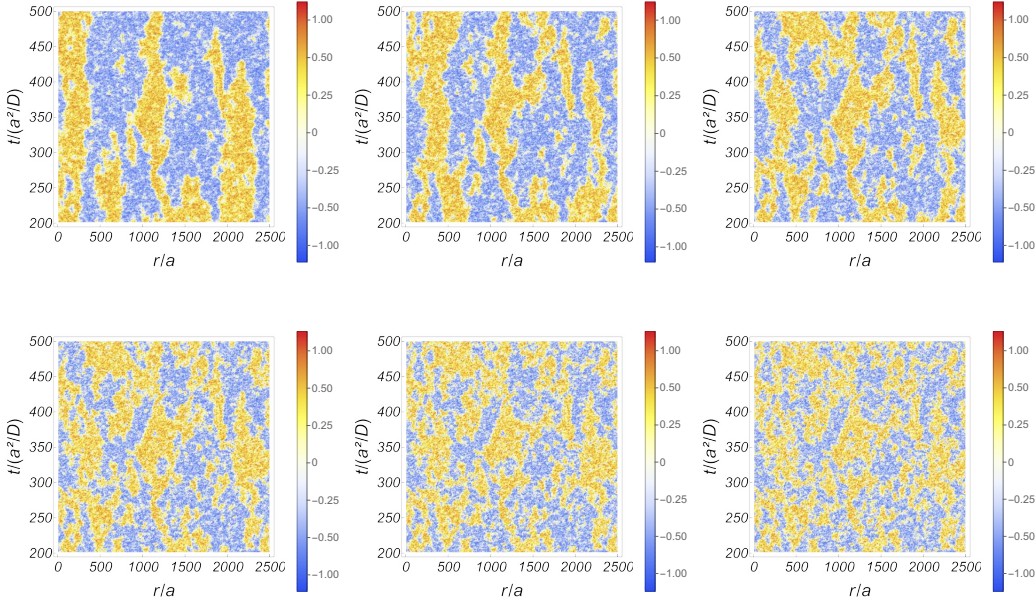

Figure 4: Local magnetization of the hydrodynamic theory, Eq. (15), at different coupling strengths $\tilde{\epsilon} = 0.1, .125, 0.15$ (first row) and $\tilde{\epsilon} = 0.175, 0.2, 0.225$ (second row), $\left(\tilde{\epsilon} = (Ja^2/D)\epsilon\right)$, as a function of space $r$ and time $t$ in units of $a$ and $a^2/D$, respectively. Parameters: $\gamma = 0.875$, $2D\chi T = 1$, $L = 10000$ where only one quarter of the system is shown. The same seed for random number generation is used for all plots.

$e^{-E_{DW}/T_{\text{eff}}}$ or

$$n_{DW} \sim e^{-c/\sqrt{\epsilon}} \quad \text{for} \ \epsilon \ll 1, \tag{18}$$

where $c = \frac{\sqrt{2D(5\gamma-4)^3}}{3a\sqrt{J}\gamma(1-\gamma)}$ if we only include fluctuations from $\xi$, ignoring corrections from $\xi_{th}$. More precisely, we use $n_{DW} \propto \epsilon^{1/4} e^{-c/\sqrt{\epsilon}}$ to fit the numerical result. The prefactor $\epsilon^{1/4}$ arises when one takes quadratic fluctuations around the optimal domain wall configuration with minimal energy into accout using that $\int e^{-c'x^2/\sqrt{\epsilon}}dx \propto \epsilon^{1/4}$. Our scaling analysis, Eqs. (14), strongly suggests that these results also hold when the second noise term $\xi^{th}$ is switched on as it has the same scaling properties. Only the prefactor $c$ should become smaller when an extra source of noise induces more domain walls. This is confirmed by our numerical results. The solid line in Fig. 5 is a fit to $\epsilon^{1/4} e^{-c/\sqrt{\epsilon}}$ at finite temperature. The fit works very well for $0.08 \lesssim \tilde{\epsilon} \lesssim 0.4$. Deviation for very small values of $\tilde{\epsilon}$ arise from finite size effects when the distance of domains $1/n_{DW}$ becomes of the order of the system size ($L = 10.000$ in our simulation). Within our numerics we obtain when including $\xi_{th}$ a value of $c = \tilde{c} \approx 0.56 \pm 0.2$ ($D = J = a = 1$) that is indeed smaller than our analytical prediction $c \approx 0.98$ obtained for the model without thermal noise. Here the error is a rough estimate obtained by using different preexponential terms ($1$, $\epsilon^{1/4}$, $a\epsilon^{1/4} + b\epsilon^{3/4}$) for the fit function. We have also performed numerical simulation where we considered only fluctuations due to $\xi$ to validate our numerical result. In this case we found a larger exponent consistent with the analytical value $c = 0.98$, see App. A.3.

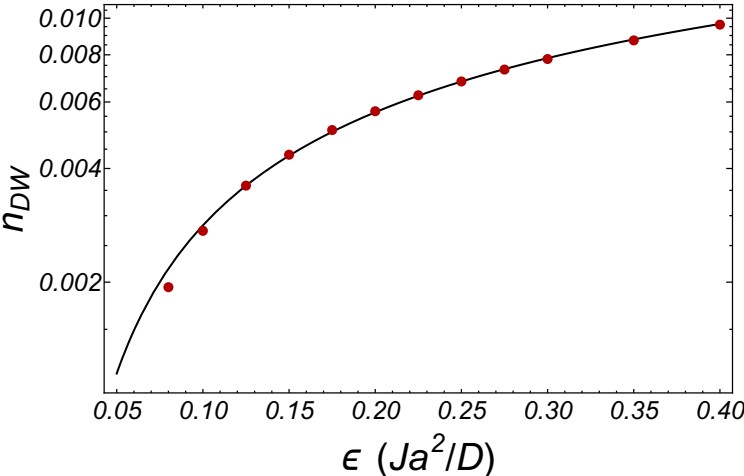

Figure 5: Domain wall density as a function of the coupling strength $\tilde{\epsilon} = (Ja^2/D)\epsilon$. The solid line shows the exponential fit $\tilde{A}\tilde{\epsilon}^{1/4}e^{-\tilde{c}/\sqrt{\tilde{\epsilon}}}$ where $\tilde{A} = (2.93 \pm 0.06) \cdot 10^{-2}$ and $\tilde{c} = 0.56 \pm 0.2$. Parameters: $L = 10000$, $\gamma = 0.875$, $2D\chi T = 1$. Results are obtained by averaging over ten noise realizations.

## 5 Discussion

Weakly driven classical and quantum systems can exhibit properties with no equilibrium analogy. Our example shows, that even a very weak driving term can induce ferromagnetism in an antiferromagnetic system. In contrast, very large Hamiltonian perturbations are needed to transform an antiferromagnet to a ferromagnet. Nevertheless, phase transitions in the driven system share many similarities with finite-temperature phase transitions, at least in cases where the stationary points of the Lindblad evolution are not noiseless absorbing dark states [31–34]. We have shown that for a weakly-driven system with approximate conservation laws one can describe the physics at large length scales by noisy hydrodynamic equations which are similar (but not identical) to corresponding equations for thermal systems. An important consequence of the noise is that phase transitions only occur in dimensions larger than one.

In the one-dimensional example analyzed by us one finds instead that at each finite noise strength a finite density of domain walls with a density proportional to $e^{-c/\sqrt{\epsilon}}$ arises. The width of the domain walls are not determined by energetic arguments but instead by the interplay of diffusion and the drive with strength $\epsilon$. Therefore the width scales with $1/\sqrt{\epsilon}$. The origin of this peculiar behavior is the (approximate) conservation of magnetization in the system, which ultimately allows one to drive the system far from equilibrium by only weak perturbations.

While we have numerically demonstrated these properties only for a simplified model with a single, bistable diffusive mode, our analytical analysis shows that these properties are generic for 1d diffusive systems where the non-equilibrium coupling to a conserved quantity (here the magnetization) drives a bistability. Our scaling and fixed-point analysis of the hydrodynamic theory of a (non-integrable) xxz chain shows, that the same type of domain walls of width $1/\sqrt{\epsilon}$ occur also when further diffusive mode exists (here: energy diffusion). As also the noise terms have the same scaling properties, the density of domain walls will follow an $e^{-c/\sqrt{\epsilon}}$ law in this case. Note, however, that the situation is different when one replaces the non-integrable xxz chain with next-nearest neighbor interactions by a model which is integrable in the absence of perturbations, $\epsilon = 0$. In this case, there is no diffusion in the uncoupled system, $\epsilon = 0$, which will necessarily lead to qualitatively different properties in the limit $\epsilon \to 0$.

Our analysis can also be seen as an example of a weakly driven system which can *not* be

described simply by a (generalized) Gibbs ensemble as used by us, e.g., in Ref. [25]. Due to the existence of several fixed points and due to strong fluctuations effects in low-dimensional systems, it is necessary to consider instead ensembles of (generalized) Gibbs ensembles with fluctuating Lagrange parameters. In more simple situations, where only a single attractive Gibbs state exists, one can instead expect that large fluctuations are sufficiently rare to allow for systematic expansions around (generalized) Gibbs states [24]. We expect that the notion of fluctuating hydrodynamics will also be very useful to explore the physics of driven approximately integrable systems with an infinite number of conservation laws.

## Acknowledgements

We acknowledge useful discussions with Sebastian Diehl, Jan Gelhausen, Zala Lenarčič and Philipp Weiß. We furthermore thank the Regional Computing Center of the University of Cologne (RRZK) for providing computing time on the DFG-funded High Performance Computing (HPC) system CHEOPS.

**Funding information** This work was supported by the DFG within CRC 1238 (project number 305 277146847, project C04) and CRC TR 183 (project number 277101999, project A01).

## A Appendix

### A.1 Generalized Forces

The generalized force $\boldsymbol{f} = (f_1, f_2)$ can to leading order in $\epsilon$ be calculated with the formula $f_i = (\epsilon/L)\langle \hat{\mathcal{L}}_1^\dagger Q_i \rangle_{r_0}^{(0)}$ ($i = 1, 2$) which yields

$$f_1 = J\frac{2\epsilon}{L}\sum_j \gamma\left(\langle P_{j-1}^\uparrow P_j^\downarrow P_{j+1}^\uparrow\rangle_{r_0}^{(0)} - \langle P_{j-1}^\uparrow P_j^\downarrow P_{j+1}^\uparrow\rangle_{r_0}^{(0)}\right) + (1-\gamma)\langle\sigma_j^z\rangle_{r_0}^{(0)},$$

$$f_2 = J^2\frac{2\Delta\epsilon}{L}\sum_j 2\gamma\left(\langle P_{j-1}^\uparrow P_j^\downarrow P_{j+1}^\uparrow\rangle_{r_0}^{(0)} + \langle P_{j-1}^\uparrow P_j^\downarrow P_{j+1}^\uparrow\rangle_{r_0}^{(0)}\right) + (1-\gamma)\langle\sigma_j^z(\sigma_{j-1}^z + \sigma_{j+1}^z)\rangle_{r_0}^{(0)}.$$

In the infinite temperature limit the force simplifies to $f_1(m) = \epsilon J\left(\frac{\gamma}{4}(1-m^2) - (1-\gamma)\right)m$.

### A.2 Noise

To calculate the noise, we follow Ref. [28] and start from the relation (called 'generalized Einstein relation' in Ref. [28])

$$\frac{d}{dt}\langle O_\alpha^\dagger O_\beta\rangle - \langle\hat{\mathcal{L}}[O_\alpha]^\dagger O_\beta + O_\alpha^\dagger\hat{\mathcal{L}}[O_\beta]\rangle = \langle\xi_\alpha^\dagger\xi_\beta\rangle$$

that can be used to calculate the noise matrix $N_{\alpha\beta} = \langle\xi_\alpha^\dagger\xi_\beta\rangle$. To do so we write the equation of motion of the operator $O_\alpha$ in the Heisenberg picture $\hat{\mathcal{L}}[O_\alpha] = i[H_0, O_\alpha] + \epsilon\hat{\mathcal{L}}_1^\dagger[O_\alpha]$. Formally, calculating the time derivative of $\langle O_\alpha^\dagger O_\beta\rangle$ yields

$$\frac{d}{dt}\langle O_\alpha^\dagger O_\beta\rangle = \langle\dot{O}_\alpha^\dagger O_\beta + O_\alpha^\dagger\dot{O}_\beta\rangle = \langle\hat{\mathcal{L}}^\dagger[O_\alpha]O_\beta + O_\alpha^\dagger\hat{\mathcal{L}}[O_\beta] + \xi_\alpha^\dagger O_\beta + O_\alpha^\dagger\xi_\beta\rangle.$$

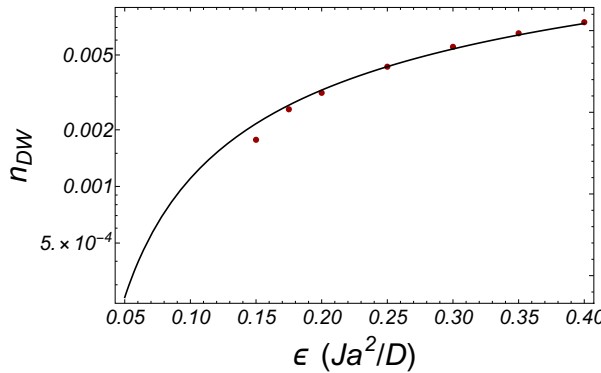

Figure 6: Domain wall density as a function of the coupling strength $\tilde{\epsilon} = (Ja^2/D)\epsilon$. The solid line shows the exponential fit $\tilde{A}\tilde{\epsilon}^{1/4}e^{-\tilde{c}/\sqrt{\tilde{\epsilon}}}$ where we obtain for fixed $\tilde{c} = 0.98$ $\tilde{A} = (4.4 \pm 0.06) \cdot 10^{-2}$. Parameters: $L = 10000$, $\gamma = 0.875$, $2D\chi T = 0$. Results are obtained by averaging over ten noise realizations.

Next we use the approximation $O_\beta(t) - O_\beta(t - \Delta t) = \int_{t-\Delta t}^{t} dt' \, \dot{O}_\beta(t')$ to write

$$\langle \xi_\alpha^\dagger(t) O_\beta(t) \rangle = \underbrace{\langle \xi_\alpha^\dagger(t) O_\beta(t - \Delta t) \rangle}_{=0} + \int_{t-\Delta t}^{t} dt' \, \underbrace{\langle \xi_\alpha^\dagger(t) \hat{\mathcal{L}}[O_\beta(t')]}_{=0, \forall t' < t} + \xi_\alpha^\dagger(t)\xi_\beta(t') \rangle$$

$$= \frac{1}{2} \int_{-\infty}^{\infty} d\tau \, \langle \xi_\alpha^\dagger(0) \xi_\beta(\tau) \rangle.$$

This finally gives

$$\frac{d}{dt} \langle O_\alpha^\dagger(t) O_\beta(t) \rangle - \langle \hat{\mathcal{L}}^\dagger[O_\alpha(t)] O_\beta(t) + O_\alpha^\dagger(t) \hat{\mathcal{L}}[O_\beta(t)] \rangle = \int_{-\infty}^{\infty} d\tau \, \langle \xi_\alpha^\dagger(0) \xi_\beta(\tau) \rangle. \tag{19}$$

We can use this relation to determine the noise-noise correlation matrix. As an example we calculate $\langle \xi_1 \xi_1 \rangle$ in the high-temperature limit which is used in the numerical simulation of our toy model. While the unitary part of the dynamics and the second Lindblad term do not yield a contribution, the first Lindblad term yields

$$\underbrace{\frac{d}{dt} \langle \sigma_j^z \sigma_j^z \rangle_{r_0}^0}_{=0} - 2\Gamma \left\langle \sum_k \left( \sigma_k^x \sigma_j^z \sigma_k^x - \sigma_j^z \right) \sigma_j^z \right\rangle_{r_0}^0 = 4\Gamma,$$

where $\Gamma = J\epsilon(1 - \gamma)$.

### A.3  Domain wall density without thermal fluctuations

Fig. 6 shows the domain wall density obtained in the absence of thermal fluctuations where the system is initially prepared in a random state. The solid line shows a fit to the function $\tilde{A}\tilde{\epsilon}^{1/4}e^{-\tilde{c}/\sqrt{\tilde{\epsilon}}}$ with the exponent $\tilde{c}$ fixed to the analytically predicted value $\tilde{c} = 0.98$. Small values of $\epsilon$ are difficult to compute due to the exponential increase in the time scale needed to obtain a steady state independent of initial conditions.

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
