# Peer review of "Bistabilities and domain walls in weakly open quantum systems"

_SciPost Physics, doi:SciPost Phys. 9, 057 (2020)_

## Round 1 · Referee Report · Anonymous (Referee 1) · 2020-8-14

Strengths

  1. Correctness within the framework of the method.
  2. Novelty.
  3. Application of an interesting method.
  4. Results apply in the thermodynamic limit.
  5. Results address a non-equilibrium situation

Weaknesses

  1. Results only concern the specifically considered model.
  2. Modeling of the coupling is a bit contrived.
  3. Results rely on some assumptions. Results can hardly be checked by another method, not even in the limit of small systems.
  4. Results are not extremely surprising.
  5. The meaning of some sentences is not clear.

Report

The paper analyzes an XXZ chain with an overnext neighhbor coupling wich is coupled via a Lindblad scheme to two baths that drag the system towards to different equilibrium states such that a NESS results. One equilibrium state has all spins aligned the other has not. The paper provides results on the occurrence of domain walls in the NESS. Methodologically the paper maps the open quantum system via hydrodynamic theory onto a set of Langevin-type equations which are then, to some extend numerically, solved.

All ingredients are timely and the theory is rather sophisticated. The results are to my best knowledge original. Thus I recommend publication, with a few changes.

Requested changes

  1. p.1: "Generalized Gibbs ensembles are powerful concepts to (approximately) describe complex interacting systems with a set of exact or approximate conserved quantities...." What precisely does this sentence mean?

  2. p.2: "Importantly this state is completely noiseless" What precisely does this sentence mean?

  3. p.8: "(they will, for example, not fulfill the second law of thermodynamics)" What precisely does this sentence mean?

  • validity: good
  • significance: ok
  • originality: high
  • clarity: good
  • formatting: excellent
  • grammar: excellent

Author:  Achim Rosch  on 2020-09-15  [id 971]

(in reply to Report 1 on 2020-08-14)

We thank the referee for the positive evaluation and for pointing out a number of unclear formulations in our text.

The referee writes:
"The paper analyzes an XXZ chain with an overnext neighhbor coupling wich is coupled via a Lindblad scheme to two baths that drag the system towards to different equilibrium states such that a NESS results. One equilibrium state has all spins aligned the other has not. The paper provides results on the occurrence of domain walls in the NESS. Methodologically the paper maps the open quantum system via hydrodynamic theory onto a set of Langevin-type equations which are then, to some extend numerically, solved. All ingredients are timely and the theory is rather sophisticated. The results are to my best knowledge original. Thus I recommend publication, with a few changes."

Below we address all questions raised by the referee:

"1. p.1: "Generalized Gibbs ensembles are powerful concepts to (approximately) describe complex interacting systems with a set of exact or approximate conserved quantities...." What precisely does this sentence mean?"

We apologize for starting our introduction with this rather technical sentence. Following the criticism by the referee, we have modified the beginning of our introduction and hope that it is more transparent now.

"2. p.2: "Importantly this state is completely noiseless" What precisely does this sentence mean?"

One way to make this statement precise is to observe that the entropy divided by the volume of the system vanishes for this state. To avoid confusion, we do not use the term noiseless in the revised version and instead only talk about the presence and absence of dark states.

" 3. p.8: "(they will, for example, not fulfill the second law of thermodynamics)" What precisely does this sentence mean?"

Again, our statement was unclear. We replaced it by “(one can, for example, continuously extract energy from the system)”.

---

## Round 1 · Referee Report · Anonymous (Referee 2) · 2020-8-26

Strengths

  • Very interesting theoretical treatment
  • Connections between equilibrium and non-equilibrium established

Weaknesses

  • Results are not many
  • Lack of a proper comparison with numerical simulations

Report

The paper is very timely and interesting, as it connects ideas from different research topics and fields. My main remark is that the developed theory is not compared with microscopic numerical simulations, either performed with ED or TN. (also, to avoid all problems with quantum systems one may just simulate a classical one). Therefore its validity and relevance are questionable.

Other comments: - It is not well explained how to go from eq 8 to eq 9, should eq 9 be intended as saddle point of the path integral? - The gradient expansion is a well known hydrodynamic concept, so maybe some reference on this may be helpful. - A derivation of 16 would be welcomed. It is obvious why $x_0 \to0$ as $D \to 0$? A free fermion system (XY chain) would display no diffusion $D=0$ so one should expect no domain wall in this case? - Appendix A.2 is very poorly written. Are the results contained in different references? If yes it should be specified. What is Einstein relation? this should be D = mu k T, nothing to do with the first equation.

Requested changes

Reply/implement the comments and if possible check the theory with a microscopic numerical simulation.

  • validity: good
  • significance: good
  • originality: high
  • clarity: high
  • formatting: excellent
  • grammar: excellent

Author:  Achim Rosch  on 2020-09-15  [id 969]

(in reply to Report 2 on 2020-08-26)

We thank the referee for the report. Below we address all questions of the referee.

The referee writes:
"The paper is very timely and interesting, as it connects ideas from different research topics and fields. My main remark is that the developed theory is not compared with microscopic numerical simulations, either performed with ED or TN. (also, to avoid all problems with quantum systems one may just simulate a classical one). Therefore its validity and relevance are questionable."

Our response:
As also pointed out by the other referee, a microscopic numerical simulation for the quantum model is completely impossible as one would have to simulate hundreds of spins. Setting up a similar microscopic classical model is in principle possible but it would still define a very challenging numerical project. While we have extensive experience in performing challenging simulations of noisy versions of Landau-Lifshitz-Gilbert equations (see, e.g., Science 340, p1076 (2013)), we do not think that one should invest a substantial amount manpower and supercomputing resources just to confirm a result which - in our opinion - has been firmly established by our paper using the combination of analytical and numerical results. As our analysis is based on a careful theoretical analysis of a problem with a well-defined small parameter, we hope that the validity and relevance of our result can be judged without further more microscopic numerical simulations. Note also that we did perform extensive numerical simulation to clarify the one question which we could not answer analytically (whether the interplay of two different types of noise terms give rise to the type of domain wall distribution known from equilibrium).
Below we address also the other comments of the referee.

The referee writes:
"It is not well explained how to go from eq 8 to eq 9, should eq 9 be intended as saddle point of the path integral?"

Our response: Eq. 9 simply follows from Eq. 8 by using the equation of motion for the density matrix, Eq. 2, and the fact that L0ρ0=0 by construction. In case that the referee instead wants to now how Eq. 8 is derived from Eq. 7, we would like to point out that we do not provide such a derivation. Our main theoretical analysis (starting from Eq. 8 and culminating in Eq. 11) is based on the Langevin approach (stochastic differential equation with a noise term) while Eq. 7 shows how a Fokker-Planck-style approach (i.e., a theory for a probability distribution) can be formulated. Within the Langevin approach one considers a single realization of the Lagrange parameter field λi(r). In this case, Eq. 8 simply states that one can locally approximate λi(r) by a constant (to leading order in a gradient approximation).

The referee writes:
"The gradient expansion is a well known hydrodynamic concept, so maybe some reference on this may be helpful. "

Our response:
Thanks for pointing out a missing reference. We decided to cite two books, the classical book on hydrodynamics by Foster and a book by Spohn, both strongly emphasizing fluctuation effects.

The referee writes:
"A derivation of 16 would be welcomed. It is obvious why x0→0 as D→0? A free fermion system (XY chain) would display no diffusion D=0 so one should expect no domain wall in this case?"

In the revised version we show explicitly which differential equation is solved by the magnetization profile. Mathematically, it is indeed obvious that the gradient term is needed to obtain a finite width. Physically, diffusions smears out the magnetization.
In a non-interacting system the diffusion constant diverges. The referee is correct that we do not expect a domain wall in this case. The whole hydrodynamic approach, on which our paper is based, breaks down in this limit.

The referee writes:
"Appendix A.2 is very poorly written. Are the results contained in different references? If yes it should be specified. What is Einstein relation? this should be D = mu k T, nothing to do with the first equation."

We apologize that we cited the relevant Ref. [28] only in the main text but not in the appendix (we have corrected that). We used the term ‘generalized Einstein relation’ as it was also used in Ref. [28]. But as we agree with the referee that this term might be misleading, we have reformulated the appendix accordingly. We hope that the revised version of the Appendix meets the expectations of the referee.

Final remark:
Finally, we would like to comment on the statement by the referee that “results are not so many”. In our opinion the main results are the following: (1) We show that arbitrarily weakly driven systems can exhibit local ferromagnetism even if the underlying model is an antiferromagnet. The weaker the couling to Lindblad operators, the stronger the tendency towards ferromagnetism due to the suppression of noise. (2) We show that energetical arguments play no role for the physics of domain walls which is instead governed by the interplay of diffusion and pumping. (3) We show that this leads to an unexpected scaling of the domain wall width and domain wall density as a function of driving strength. (4) We show that fluctuating hydrodynamics is a powerful tool to address such questions. (5) We provide an explicit counterexample for the (naïve) use of Gibbs ensembles in weakly driven systems.

---

## Round 1 · Referee Report · Anonymous (Referee 3) · 2020-8-29

Strengths

The manuscript provides an interesting analysis of the hydrodynamic description of a weakly open bistable quantum system.

Weaknesses

The authors start off from the 1D XXZ chain (first unnumbered equation in section 2), but then they really investigate the toy model Eq. (15) for the fluctuating domain walls. I understand that they introduce this toy model because it remains a major challenge to compute the parameters of the full hydrodynamic limit Eq. (11).

One consequence of this order of steps is that the authors should discuss to which extent the conclusions they derive from the toy model are general, or at least apply to the original problem. The last sentence of the abstract briefly hints at this issue, but related comments are absent in the concluding section 4 "Discussion".

Secondly, the toy model only has one conserved quantity such that the connection to the generalized Gibbs ensemble is lost that is the starting point of this work (first sentence of abstract and first words of the Introduction).

Report

The manuscript contains some interesting ideas, but in the present form, it makes the impression of an addendum to previous work by the same authors, Refs. [10,24] of the manuscript. Thus, with some minor changes listed below, it may be suitable for publication in SciPost Physics Core, but in order to meet the standards of the flagship SciPost Physics, more work would probably be required.

Requested changes

1- The comment "The emergence of bistabilites in non-equilibrium systems in the limit of strong drive and dissipation has recently gained increased attention ..." at the top of page 3 is a bit strange given that none of the related references [12-19] dates from after 2017. I recommend to remove at least the "recently".

2- There are $P$s in Eq. (6) and in appendix A that seem to serve the same purpose whereas the letter $P$ is also used in and below Eq. (7). In the latter instance, $P$ is explained to be a probability functional while in the former case, one can only guess that these are projection operators. I think that a proper definition should be added.

3- The $e$ and $m$ that appear with Fig. 2 remain a bit obscure . An explanation of $m$ as magnetization (density) is hidden in the text while in the case of $e$ one can only guess that it is energy density. I think that an explanation of this notation should be added, preferably without hiding it in the middle of the text.

4- The authors should explain the meaning of the triangles and their colors in Fig. 2 (at least I did not find an explanation).

5- The legend of Fig. 3 is too small to be easily readable. In addition, in the current form, one only sees the result for Eq. (16) in the main panel of Fig. 3 while in its inset one sees only numerical findings. If one is supposed to see a data collapse in the main panel, the authors should make an effort to render the numerical data visible. Otherwise, one should assign the legend for the numerical data to the inset and not the main panel (and maybe swap main panel and inset).

6- The structure of the work might be more accessible if the simplified toy model Eq. (15) and its analysis were contained in a separate section, i.e., an additional section header somewhere in the paragraph preceding Eq. (15) might be helpful.

7- If I understand correctly, "simplified toy model" would be a more precise characterization of Fig. 4 than the present "system" in its caption.

8- The assignment of the different values of $\tilde{\epsilon}$ to the different panels of Fig. 4 is not completely clear. I think that the authors should either label the panels or explain the order in words.

9- In the references, there are some typographic errors. In particular, "Phys. Rev. A" in Ref. [1] should read "Nature" and the authors should correct the spelling of their collaborator in Ref. [10].

  • validity: good
  • significance: good
  • originality: good
  • clarity: ok
  • formatting: excellent
  • grammar: excellent

Author:  Achim Rosch  on 2020-09-15  [id 970]

(in reply to Report 3 on 2020-08-29)

We thank the referee for the report and the careful reading of our manuscript. The report helped us substantially to improve the manuscript.
Below we comment on the list of weaknesses pointed out by the referees and all other remarks of the author.

The referee writes:
"The authors start off from the 1D XXZ chain (first unnumbered equation in section 2), but then they really investigate the toy model Eq. (15) for the fluctuating domain walls. I understand that they introduce this toy model because it remains a major challenge to compute the parameters of the full hydrodynamic limit Eq. (11). One consequence of this order of steps is that the authors should discuss to which extent the conclusions they derive from the toy model are general, or at least apply to the original problem. The last sentence of the abstract briefly hints at this issue, but related comments are absent in the concluding section 4 "Discussion". Secondly, the toy model only has one conserved quantity such that the connection to the generalized Gibbs ensemble is lost that is the starting point of this work (first sentence of abstract and first words of the Introduction)."

Our reply:
We thank the referee for pointing out that a central claim of our paper, namely that all qualitative results of the toy model fully apply both to the original model and also to a wide class of similar models, has apparently not been emphasized and substantiated sufficiently strongly in the previous version of the paper. We have therefore made two important changes. First, in the new Sec. 4 we added below Eq. 15 a new paragraph “Formally, the use….” which gives the precise argument why the effective order parameter theory is justified. One argument is, for example, that the omitted mode actually obtains a mass of order 1 in the properly rescaled theory due to the coupling to the Lindblad operators. Furthermore, we added a paragraph “While we have…” to the conclusions which discusses the broader validity of the result (but also points out that it does not apply to integrable models).

In the report the referee furthermore states:
"The manuscript contains some interesting ideas, but in the present form, it makes the impression of an addendum to previous work by the same authors, Refs. [10,24] of the manuscript. Thus, with some minor changes listed below, it may be suitable for publication in SciPost Physics Core, but in order to meet the standards of the flagship SciPost Physics, more work would probably be required."

Our reply:
While part of the motivation of our work came from our previous publications, none of the basic physics of the present paper (phase transitions and bistabilities) was even mentioned in one of our previous papers. We do not think that one can argue that the theory of phase transitions in equilibrium is an “addendum” to the theory of thermodynamics. Similarly, our theory of phase transitions/bistabilities in weakly driven systems should not be considered as an addendum to our previous work. Some main results are the following: (i) We show how arbitarily weak drive can transform an antiferromagnet locally into a ferromagnet, (ii) we find a new type of domain walls whose width is not governed by energetics but the interplay of drive&diffusion, (iii) we obtain an unexpected scaling for domain width and domain distance with Lindblad coupling, (iv) we argue in the revised versions why these results are generic. And finally,
the present work does construct (v) an explicit counterexample which show that (generalized) Gibbs ensembles as used previously by us (and many others) can fail in the case of bistabilities in low-dimensional systems.

Below we comment on all other issues raised by the referee:
"1- The comment "The emergence of bistabilites in non-equilibrium systems in the limit of strong drive and dissipation has recently gained increased attention ..." at the top of page 3 is a bit strange given that none of the related references [12-19] dates from after 2017. I recommend to remove at least the "recently"."

Indeed, most of the references which we cite here are from 2017. We have reformulated the sentence.

"2- There are Ps in Eq. (6) and in appendix A that seem to serve the same purpose whereas the letter P is also used in and below Eq. (7). In the latter instance, P is explained to be a probability functional while in the former case, one can only guess that these are projection operators. I think that a proper definition should be added."

We added the missing definition.

" 3- The e and m that appear with Fig. 2 remain a bit obscure . An explanation of m as magnetization (density) is hidden in the text while in the case of e one can only guess that it is energy density. I think that an explanation of this notation should be added, preferably without hiding it in the middle of the text."

We agree that it is useful to repeat the definition also in the figure caption as we do in the revised version.

"*4- The authors should explain the meaning of the triangles and their colors in Fig. 2 (at least I did not find an explanation)."

The triangles are actually arrows denoting the direction (and via the color) the amplitude of the force. We improved the figure & caption.

"5- The legend of Fig. 3 is too small to be easily readable. In addition, in the current form, one only sees the result for Eq. (16) in the main panel of Fig. 3 while in its inset one sees only numerical findings. If one is supposed to see a data collapse in the main panel, the authors should make an effort to render the numerical data visible. Otherwise, one should assign the legend for the numerical data to the inset and not the main panel (and maybe swap main panel and inset)."

We modified the figure following the suggestion of the referee.

"6- The structure of the work might be more accessible if the simplified toy model Eq. (15) and its analysis were contained in a separate section, i.e., an additional section header somewhere in the paragraph preceding Eq. (15) might be helpful."

We are following the suggestion of the referee by adding Sec. 4.

" 7- If I understand correctly, "simplified toy model" would be a more precise characterization of Fig. 4 than the present "system" in its caption."

We state now more precisely which system we are referring to. Note, however, that our “toy model” is a highly non-trivial nonlinear and noisy field theory.

"8- The assignment of the different values of ~ϵ to the different panels of Fig. 4 is not completely clear. I think that the authors should either label the panels or explain the order in words."

Changed.

"9- In the references, there are some typographic errors. In particular, "Phys. Rev. A" in Ref. [1] should read "Nature" and the authors should correct the spelling of their collaborator in Ref. [10]."

Thanks for spotting these typos.

---

## Round 3 · Referee Report · Anonymous (Referee 5) · 2020-9-18

Report

The revised version of the manuscript has improved significantly. In particular, all concrete issues from my first report have been addressed.

I would furthermore like to thank the authors for the clarifications not only via their revisions, but also their replies. Thanks to them, I recommend publication of the manuscript in its present form.

Requested changes

Pay attention to placement of figures (in particular Figs. 1 and 3) during production.

---

## Round 3 · Referee Report · Anonymous (Referee 4) · 2020-9-18

Report

The authors addressed all comments and concerns in a reasonable manner. Thus I recommend publication of the paper as it is.

---

## Round 3 · Referee Report · Anonymous (Referee 6) · 2020-10-5

Report

I thank the authors to reply to all points and to carefully address them. I recommend publication of the manuscript in the present form.

---

## Round 3 · Author Response

Please find attached our resubmission to SciPost Physics.

---

## Round 3 · List of Changes

In reply to the referees, we made a number of changes. The most important ones are the following:

1) p.8, a new paragraph "Formally, ...." has been added providing a formal justification why the simplified hydrodynamic equations
will reproduced the qualitative features of the full hydrodynamic approach
2) p.11, a new paragraph "While we..." in the conclusion also emphasized that the results apply to a large class of problems
3) p. 10, below Eq. (18), we improved the fitting procedure and added error bars to the results. The new appendix A.3 and Fig. 6 shows a similar analysis of the data obtained by omitting non-thermal noise.
4) Introductory sentence on p. 1 has been reformulated. Similarly, statements on dark states and on the 2nd law of thermodynamics on p . 2 and 8 have been reformulated.
5) Ref. [26] and [27] have been added
6) minor changes to Fig. 2 and Fig. 3 (data is not modified)
7) p. 7, a new header "Simplified hydrodynamic model: order parameter theory" has been used to structure the paper in a better way.

---

## Editorial Decision

published